# Effect of Tiotropium Soft Mist Inhalers on Dynamic Changes in Lung Mechanics of Patients with Chronic Obstructive Pulmonary Disease Receiving Mechanical Ventilation: A Prospective Pilot Study

**DOI:** 10.3390/pharmaceutics13010051

**Published:** 2020-12-31

**Authors:** Pin-Kuei Fu, Yu-Feng Wei, Chau-Chyun Sheu, Chen-Yu Wang, Chi-Kuei Hsu, Chia-Min Chen, Wei-Chih Chen, Kuang-Yao Yang

**Affiliations:** 1Department of Critical Care Medicine, Taichung Veterans General Hospital, Taichung 40705, Taiwan; yetquen@gmail.com (P.-K.F.); chestmen@gmail.com (C.-Y.W.); 2College of Human Science and Social Innovation, Hungkuang University, Taichung 43302, Taiwan; 3Department of Computer Science, Tunghai University, Taichung 40705, Taiwan; 4Division of Pulmonary Medicine, Department of Internal Medicine, E-Da Hospital/I-Shou University, Kaohsiung 82445, Taiwan; yufeng528@gmail.com (Y.-F.W.); ospreyhsu@gmail.com (C.-K.H.); 5School of Medicine for International Students, College of Medicine, I-Shou University, Kaohsiung 82445, Taiwan; 6Division of Pulmonary and Critical Care Medicine, Department of Internal Medicine, Kaohsiung Medical University, Kaohsiung 80756, Taiwan; sheucc@gmail.com (C.-C.S.); kmuronald@gmail.com (C.-M.C.); 7Department of Internal Medicine, School of Medicine, College of Medicine, Kaohsiung Medical University, Kaohsiung 80708, Taiwan; 8Department of Nursing, Hungkuang University, Taichung 43302, Taiwan; 9Department of Chest Medicine, Taipei Veterans General Hospital, Taipei 11217, Taiwan; wiji.chen@gmail.com; 10Faculty of Medicine, School of Medicine, National Yang-Ming University, Taipei 11217, Taiwan; 11Institute of Emergency and Critical Care Medicine, School of Medicine, National Yang-Ming University, Taipei 11217, Taiwan; 12Cancer Progression Research Center, National Yang-Ming University, Taipei 11217, Taiwan

**Keywords:** COPD, tiotropium, soft mist inhalers, long-acting muscarinic antagonist, mechanical ventilation, lung mechanics

## Abstract

The effects of tiotropium bromide soft mist inhalers (SMIs) in patients with chronic obstructive pulmonary disease (COPD) receiving mechanical ventilation remain unexplored. This study investigated the dynamic effects of a tiotropium SMI on lung mechanics and gas exchange in these patients. We analyzed 11 mechanically ventilated and hemodynamically stable patients with COPD who experienced acute exacerbation and were ready to be weaned from the ventilator. Two puffs of tiotropium (2.5 μg/puff) were administered with a T-adaptor connected to the ventilator circuit. Lung mechanics—peak inspiratory pressure, plateau pressure, mean airway pressure, maximum respiratory resistance (Rrs), and gas exchange function—were analyzed. The two-puff tiotropium SMI treatment led to the greatest reduction in Rrs at 6 h, with the Rrs returning to baseline gradually, and significantly improved the PaO_2_/FiO_2_ ratio at 24 h. Compared with baseline values, tiotropium SMI had the strongest effect on Rrs between hours 3 and 6 but did not significantly affect hemodynamic parameters. Tiotropium SMI administration in mechanically ventilated patients with COPD achieved the greatest reduction in Rrs at 6 h and significantly improved the PaO_2_/FiO_2_ ratio at 24 h. Future studies should investigate whether the bronchodilation effect can be improved with increased dosage or frequency.

## 1. Introduction

Chronic obstructive pulmonary disease (COPD), a chronic inflammatory disease of the lower respiratory tract, is the third leading cause of death worldwide in 2020. In 2017, COPD was the cause of 3.2 million deaths globally [1,2]. Acute exacerbation of COPD (AECOPD) commonly leads to hospitalization and may progress to respiratory failure requiring ventilatory support [2,3]. Patients with AECOPD receiving mechanical ventilation are commonly treated with short-acting bronchodilators delivered with nebulizers [4]. Metered-dose inhalers (MDIs) using a compatible spacer device have been shown to be as effective as nebulizers [5,6,7]. Long-acting bronchodilators, including tiotropium, are commonly used in patients with stable COPD to improve their respiratory symptoms and prevent another episode of AECOPD [8].

Tiotropium bromide is a long-acting muscarinic agent (LAMA). A soft mist inhaler (SMI) is a propellant-free multidose inhaler that uses mechanical power from a spring to deliver a metered dose of medication as a fine, slow-moving, long-lasting soft mist [9,10]. A tiotropium bromide SMI is employed as a maintenance bronchodilator to relieve airflow limitation and the symptoms of COPD [11,12]. The quality of dose and particle size distribution are independent of the patient’s inspiratory flow rate [13]. However, tiotropium bromide SMIs are designed for spontaneously breathing patients and not yet recommended as a standard treatment for patients with COPD receiving mechanical ventilatory support. Currently, no LAMA is available for patients with COPD who are intubated, even after the acute stage has subsided. The current Global Initiative for Chronic Obstructive Lung Disease (GOLD) guidelines recommend initiating maintenance therapy with long-acting bronchodilators as soon as possible before discharge to lower the risk of AECOPD [14,15].

LAMA is often prescribed by physicians as a maintenance therapy for patients with COPD; however, the efficacy of pulmonary drug deposition and optimal method of connecting an SMI to the ventilator system remain unclear [16,17]. An animal model study indicated that tiotropium inhalation may attenuate pulmonary inflammation induced by severe airway obstruction, potentially benefitting patients with AECOPD [18]. Various methods have been attempted for connecting the SMI to the ventilator circuit, including “an SMI-specific in-line adapter” or “a T-adapter” [19,20,21]. No study has analyzed the effects of a tiotropium SMI in mechanically ventilated patients with AECOPD or the efficiency of administering an SMI with a T-adaptor in humans. The present multicenter study examined the duration of the bronchodilator effect of tiotropium and consequent dynamic changes in lung mechanics. Tiotropium was administered using a SMI and adapter in hemodynamically stable patients with COPD receiving mechanical ventilation and who were not ready to be extubated.

## 2. Materials and Methods

### 2.1. Study Design and Patients

This prospective pilot cohort study was conducted at the intensive care units of three medical centers in North, Central, and South Taiwan. From January 2018 to May 2019, 14 patients receiving mechanical ventilation, aged older than 40 years, and experiencing AECOPD were enrolled. COPD was defined according to the Global Initiative for Chronic Obstructive Lung Disease (GOLD) guidelines [11]. Patients enrolled in the study were in a stable condition, which means that their AECOPD was under control but they were not ready to be weaned from ventilatory support. Stable condition was defined as oxygen support with fraction concentration of inspired oxygen (FiO_2_) ≤ 0.4, positive end-expiratory pressure (PEEP) ≤ 5 cm H_2_O, absence of fever (temperature measured using an ear thermometer, <38 °C), and no vasopressor or inotropic agent use for more than 24 h. The exclusion criteria were as follows: presence or history of other chronic respiratory tract disorders such as asthma, interstitial lung disease, bronchiectasis, and lung cancer; clinically significant medical disorders such as severe congestive heart failure (New York Heart Association Functional Classification III–V), liver cirrhosis (Child–Pugh score C), and chronic kidney disease requiring regular hemodialysis; and use of long-acting bronchodilators or systemic corticosteroid therapy (>1 mg/kg/day of prednisolone or equivalent) for >24 h. The study protocol was approved by the Institutional Review Board (IRB) of four hospitals (IRB number, CE17323B; approval date, 14 December 2017, and Taipei Veterans General Hospital IRB No. 2018-04-008CC; approval date, 16 May 2018). Written informed consent was obtained from all participants or their authorized representatives before enrollment.

### 2.2. Protocol and Lung Mechanics Measurement

A soft mist inhaler (SMI) is a propellant-free multidose inhaler that uses mechanical power from a spring to deliver a metered dose of medication as a fine, slow-moving, long-lasting soft mist [9,10]. Two puffs of a tiotropium SMI (tiotropium, 2.5 μg/puff) were administered to patients with COPD receiving mechanical ventilation by using an adapter at 9:00 am per the GOLD guidelines [12,22]. The tiotropium SMI was connected to the ventilator circuit at 15 cm from the Y-piece with a T-adaptor, as reported [16,17]. After two puff of Tiotropium SMI actuation was performed at the initiation of the inspiration phase of the volume control mode, the lung mechanism will be checked according to the time points of protocol at 1 h, 3 h, 6 h, 12 h, and 24 h. To maintain airway hygiene and oxygenation (FiO_2_ = 1.0), an open method of sputum suction was performed at least 5 min before initiation of the protocol and before each measurement, as described in our previous report [23]. The 5-min interval was sufficient to prevent transient bronchoconstriction after sputum suction [24]. During the measurement of lung mechanics, the patient was sedated with an intravenous short-acting anesthetic (midazolam or propofol) to ensure a Richmond Agitation–Sedation Scale score between −3 and −4. The ventilator setting was maintained in the volume-control mode with tidal volume ≤ 8 mL/kg, flow rate of 50–60 L/min, FiO_2_ ≤ 0.4, and PEEP ≤ 5 cmH2O. Parameters of lung mechanics were peak inspiratory pressure (PIP), mean airway pressure (Pmean), plateau pressure (Pplat), and maximum resistance of the respiratory system (Rrs). The parameters were measured before tiotropium SMI introduction (baseline) and at 1 h, 3 h, 6 h, 12 h, and 24 h. Lung mechanics were measured using the monitor and program setting of the mechanical ventilator (SERVO 300 or SERVO-I, Siemens, Munich, Germany). The occlusion method was used to measure lung mechanics, as previously reported [25,26]. Gas exchange function—partial pressure of oxygen in arterial blood (PaO_2_), FiO_2_, and PaO_2_/FiO_2_ ratio (PF ratio)—were recorded at baseline and at 3 h and 24 h. Physiological parameters—heart rate (HR) and mean artery pressure (MAP)—were monitored and recorded to evaluate the effects on hemodynamic status at three time points: 30 min before tiotropium SMI actuation and at 3 h and 24 h after.

### 2.3. Statistical Analysis

Statistical analysis was performed using SPSS v22.0 (IBM, Armonk, NY, USA). Respiratory parameters were analyzed using the nonparametric Kruskal–Wallis test to determine the differences between various time points. The Dunn–Bonferroni test was performed for multiple comparisons of nonparametric pairwise independent groups. A *p* value < 0.05 was considered significant for all tests.

## 3. Results

### 3.1. Patients’ Clinical and Demographic Characteristics

Fourteen patients with COPD meeting the inclusion criteria were enrolled. Of them, one had new onset of fever, one developed coffee ground vomitus from the nasogastric tube, and one withdrew from the study; thus, 11 patients who completed the whole protocol were considered in the analysis. Their median age was 77 years (interquartile range (IQR): 67–83 years), they were predominately male (9 patients, 81.8%), and they were classified into GOLD groups B (4 patients, 36.4%) and D (7 patients, 63.6%; Table 1).

### 3.2. Lung Mechanics at Different Time Points

Table 2 presents the median values of various lung mechanics parameters at different time points. No significant difference was noted in multiple comparisons among the median values at each time point (*p* > 0.05). However, compared with baseline, the PIP, Pmean, and Pplat were higher at 1 h, 3 h, and 6 h, respectively (Figure 1). Although nonsignificantly different compared with baseline, Rrs reached its lowest value at 6 h post administration and had gradually returned to the baseline value by 12 h (Figure 1).

### 3.3. Physiological Parameters at Different Time Points

As presented in Table 3, no significant differences were discovered among the three time points in the physiological parameters HR and MAP (all *p* > 0.05). Compared with baseline, the difference in the changes in HR and MAP was smaller than 10% (Figure 2).

### 3.4. Blood Gas Parameters and PaO_2_/FiO_2_ at Different Time Points

Arterial blood gas parameters—PH, PaO_2_, PaCO_2_, HCO_3−_, base excess (BE), and FiO_2_—were monitored and recorded 30 min before and at 3 and 24 h after tiotropium SMI actuation (Table 3). As presented in Table 3, no significant differences were observed among the three time points in the blood gas parameters PH, PaO2, PaCO2, HCO3^−^, BE, and FiO_2_ (all *p* > 0.05). The median PF ratio was 191 (IQR: 175–221) at baseline, 221.75 (IQR: 202–292) at 3 h, and 246.67 (IQR: 211.25–324) at 24 h, indicating significant improvement over time (*p* = 0.012). The Dunn–Bonferroni pairwise comparison test revealed that the PF ratio was significantly higher (by 23%) at 24 h compared with that at baseline (*p* = 0.009).

## 4. Discussion

This pilot study had three major findings. First, the Rrs at 6 h was more than 10% lower than the baseline value. Second, the effects of the standard dose of the tiotropium SMI (2.5 μg/puff, 2 puffs/day) did not last for 24 h in the patients with COPD receiving mechanical ventilation. Third, administration of the tiotropium SMI in patients with COPD receiving mechanical ventilation did not have adverse effects on hemodynamic parameters but improved the oxygenation status (PF ratio) at 24 h. To the best of our knowledge, this is the first real-world study to examine the effect of a tiotropium SMI on lung mechanics in patients with COPD receiving mechanical ventilation.

Only a few studies have discussed the delivery method for a tiotropium SMI in patients receiving mechanical ventilation [20,21]. A tiotropium bromide SMI is prescribed as a maintenance bronchodilator to relieve airway obstruction and the symptoms of COPD [11,12]. The quality of dose and particle size distribution are independent of the patient’s inspiratory flow rate [13]. Although an in-line adapter for SMIs was approved by the US Food and Drug Administration (FDA) in April 2020, is commercially available, and provides an access port to facilitate tiotropium administration in such patients [27], the most common approach has been to connect to a T-adapter using a silicon adapter at 15 cm from the Y-piece and 90° from the inspiratory limb of the ventilator [11,16,19]. Ke et al. stated that the location resulting in the highest inhaled dose from a tiotropium SMI was in the inspiratory limb at 15 cm from the Y-piece [16]. Therefore, we used this method in the current study to administer the highest-possible delivered dose.

Another critical issue is the optimal dose delivered by the tiotropium SMI for patients receiving mechanical ventilation. In an in vitro study, Ke et al. reported eight actuations of a tiotropium SMI into the ventilated circuit, and the delivery efficiency was found to be only 22.9 ± 5.8% [16], whereas for Fang et al., who used four actuations, it was only 1.02 ± 0.58% [17]. The dosage of the tiotropium SMI in these two studies was double (four puffs) or four times (eight puffs) higher than the standard dose (two puffs) for spontaneously breathing patients. As shown in these two in vitro studies, the inhaled tiotropium dose with simulated spontaneous breathing was nearly 20 times greater than the dose administered through a T-adapter in an endotracheal tube (22.9 ± 5.8% and 1.02 ± 0.58%, respectively; *p* < 0.001) [16]. Even when tiotropium was actuated from the FDA-approved in-line adaptor, the delivered dosage was one third that for simulated spontaneous breathing (7.68 ± 0.98% vs. 22.08 ± 4.8%, respectively; *p* < 0.001) [16]. In fact, studies evaluating the use of pressurized MDIs (pMDIs) in patients receiving mechanical ventilation have observed that significant bronchodilator effects can be achieved using a pMDI with as few as four puffs [28,29]. Therefore, the dosage of a tiotropium SMI for mechanically ventilated patients should be at least four times higher than that for spontaneously breathing patients when the SMI is connected to a T-adaptor and at least three to four times higher when the SMI is connected using the in-line adaptor.

In our study, although inter-time-point differences in the parameters of lung mechanics did not reach statistical significance, we observed trends of a decrease in PIP, Pmean, and Pplat at 1 h, 3 h, and 6 h after two SMI administrations compared with the baseline values. However, the decreased values gradually returned to the baseline values after the 6 h time point. This finding is the same as that in our previous study evaluating the effects of salmeterol and fluticasone on lung mechanics in patients receiving mechanical ventilation, which revealed that salmeterol has the maximum bronchodilation effect, causing 15% lower PIP than the baseline value [23]. Malliotakis et al. evaluated the duration of salmeterol-induced bronchodilation and noted that salmeterol led to an approximately 10% reduction in PIP at 2 to 6 h but that the PIP gradually returned to the baseline after the 6 h time point [7,30]. Other major parameters evaluated in the current study—such as Pmean, Pplat, and Rrs—reached their minimum at 6 h after tiotropium SMI actuation but had returned to the baseline value by 12 h. The greatest change was observed in Rrs, which was decreased by approximately 13% at 6 h. The poor performance of long-acting bronchodilators in mechanically ventilated patients compared with that in spontaneously breathing patients may be due to multiple factors such as use of a chamber with MDIs, connection of the inhalation device to the inspiratory line at 10–30 cm from the endotracheal tube, humidity in the ventilator circuit, the tidal volume setting, the inspiration flow rate, the respiratory frequency, and use of a breath-holding maneuver after inhalation [5,23,29]. Although several studies have measured lung mechanics after administering long-acting bronchodilators [5,16,17,23,30,31], our findings imply that the dose to the patients with COPD receiving mechanical ventilation was insufficient due to the limitations of the mechanical ventilated circuit and setting.

We observed a significantly increased PF ratio after tiotropium SMI actuation: 191.0 cm H_2_O increased to 221.75 cm H_2_O at 3 h and 246.67 cm H_2_O at 24 h (*p* = 0.012). Introducing inhaled-adrenergic agents to patients with obstructive airway disease often results in a transient decrease in PaO_2_ due to pulmonary vasodilation [32,33]. By contrast, the administration of anticholinergic bronchodilators has relatively weak effects on PaO_2_ [34], and tiotropium may even slightly increase PaO_2_ [35,36]. Tiotropium can improve nocturnal oxygenation in patients with COPD by reducing the cholinergic tone to relieve bronchial obstruction and consequently improve the ventilation-to-perfusion ratio [37,38]. However, the increased PF ratio in the current study is not only due to the maximum bronchodilation effect of tiotropium at 6 h, but also may have been due to multiple factors, including intensive chest care and sputum suction, the anticholinergic effect of tiotropium, and improved ventilation–perfusion mismatch.

This pilot study has several limitations. First, our sample was small. We enrolled 14 patients from three medical centers in Taiwan, but recruiting patients with COPD who have just recovered from acute bronchospasm but are not yet ready to be weaned from ventilatory support is difficult, partly because of the small time window. Another barrier to enrollment was the need for deep sedation to measure lung mechanics; this sedation is not easily accepted by patients or their family. Previous studies investigating the effects of long-acting bronchodilators on lung mechanics in patients with COPD receiving ventilation have also enrolled 10 or fewer cases due to the aforementioned limitations [7,23,30]. Second, we connected the tiotropium SMI to the ventilator circuit at 15 cm from the Y-piece by using a T-adaptor and not a commercial in-line adaptor for SMIs; the commercial in-line adaptor only recently approved by the US FDA [27] was not available in Taiwan when this research was conducted. Two recent in vitro studies from Taiwan have used the same method as ours to test the delivery efficacy of a tiotropium SMI during mechanical ventilation [16,17]. Third, the baseline lung condition of the enrolled patients may have been heterogeneous, meaning that some enrollees may have had different COPD phenotypes, such as bronchitis or emphysema. Finally, our results may not be generalizable to all patients with COPD receiving mechanical ventilation, because this study included only hemodynamically stable patients with AECOPD and without clear pneumonia patches who were not ready to be weaned from ventilatory support. Further studies are required to evaluate the effects of different dosages of a tiotropium SMI on lung mechanics when the SMI is connected using the now FDA-approved commercial in-line adaptor; additionally, a more diverse group of patients with COPD receiving mechanical ventilation should be recruited.

## 5. Conclusions

Tiotropium SMI inhalation therapy in patients with COPD receiving mechanical ventilation resulted in the greatest reduction in Rrs at 6 h post actuation and significantly improved the oxygenation index PaO_2_/FiO_2_ at 24 h. The greatest change of lung mechanisms by Tiotropium SMI inhalation therapy at 6 h was observed in the reduction of Rrs, PIP, Pmean, and Pplat compared with the baseline values. These changes of lung mechanisms may contribute from the bronchodilation effect of Tiotropium to reduce the hyperinflation of COPD. Increased dose or frequency may improve the bronchodilation effect of tiotropium SMI inhalation therapy in ventilated patients with COPD.

## Figures and Tables

**Figure 1 pharmaceutics-13-00051-f001:**
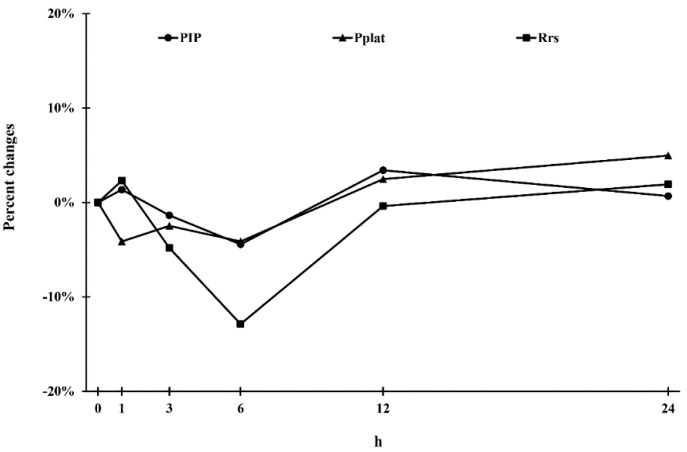
Percent changes (△-difference %) in Peak inspiratory pressure (PIP), mean airway pressure (Pmean), plateau pressure (Pplat), and maximum resistance of the respiratory system (Rrs).

**Figure 2 pharmaceutics-13-00051-f002:**
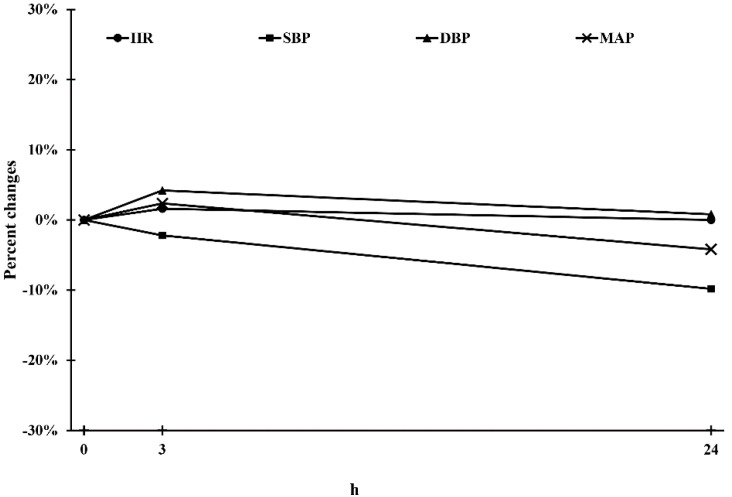
Physiology parameters during the observational period. HR: heart rate; SBP: systolic blood pressure; DBP: diastolic blood pressure; MAP: mean arterial pressure.

**Table 1 pharmaceutics-13-00051-t001:** Patient characteristics.

Characteristics	Median	IQR
Age	77.00	(67–83)
Gender (*n*, %) Male	9	(81.8%)
GOLD (*n*, %)		
1–2 (FEV1 ≧ 50%)	5	(45.5%)
3–4 (FEV1 < 50%)	6	(54.5%)
Group (*n*, %)		
B	4	(36.4%)
D	7	(63.6%)
Comorbidities (*n*, %)		
Cardiovascular disease	6	(42.9%)
Diabetes Mellitus	3	(21.4%)
Chronic liver disease	1	(7.1%)
Chronic renal disease	1	(7.1%)
Chronic neurogenic disease	2	(14.3%)
SOFA score	5.0	(4–8)
APACHE-Ⅱ score	21.0	(17.5–25.0)
Pulmonary Function test		
FEV1 (L)	1.13	(0.71–1.55)
FEV1(%)	43	(28–75)
FEV1/FVC (%)	54	(44–70)
Vital signs (at baseline)		
HR (beats/min)	74	(57–103)
MAP (mmHg)	76	(61–84)
Arterial blood gas (at baseline)		
PH	7.45	(7.38–7.46)
PaO_2_	74.4	(70–82.6)
HCO_3_^−^	27.4	(22.8–30.6)
PaCO_2_	37.9	(31.8–52.3)
PaO_2_/FiO_2_	191.0	(175–221)

GOLD: Global Initiative for Chronic Obstructive Lung Disease; SOFA: Sequential Organ Failure Assessment; APACHE II: Acute Physiology and Chronic Health Evaluation II; FEV1: The forced expiratory volume in one second; FVC: Forced vital capacity; HR: Heart rate; MAP: Mean arterial pressure. Patient was classified as group B (more symptoms and less exacerbation) and D (more symptoms and more frequently of exacerbation) [2].

**Table 2 pharmaceutics-13-00051-t002:** Respiratory parameters from ventilator.

Parameters	Zero	1st h	3rd h	6th h	12th h	24th h	*p*
PIP	24.0 (18–36)	24.0 (19–39)	26.0 (20–30)	22.0 (18–32)	27.0 (19–35)	25.0 (21–33)	0.775
Pplat	15.50 (13–17)	14.0 (13–16)	15.0 (14–15.8)	15.0 (13.3–15.8)	15.0 (14–16.8)	15.5 (14.3–17.5)	0.680
Pmean	9.00 (9–11)	9.0 (8–10)	9.0 (8–10)	9.0 (7–11)	9.0 (8–11)	9.0 (9–11)	0.864
PEEP	5.00 (4.25–5)	5.0 (4.25–5)	5.0 (5–5)	5.0 (5–5)	5.0 (4.25–5)	5.0 (4.25–5)	0.940
RR	16.00 (14–18)	15.0 (14–18)	16.0 (14–18)	14.0 (14–20)	15.0 (14–16)	14.0 (14–15)	0.132
MVi	7.65 (6.88–8.05)	6.85 (5.7–7.93)	7.3 (6–8.7)	6.65 (5.85–8.45)	7.1 (5.9–8.28)	7.65 (6.25–7.98)	0.069
MVe	7.70 (7–8.4)	7.40 (5.7–8.6)	7.6 (6.1–9)	7.0 (6.1–9.1)	7.4 (6.2–8.2)	7.6 (6.2–7.9)	0.247
VTi	472.0 (445–505)	470.0 (445–498)	471.0 (446–503)	468.0 (446–498)	474.0 (445–518)	477.0 (446–545)	0.150
VTe	467.0 (432–516)	475.0 (432–477)	463.0 (405–474)	469.0 (422–497)	483.0 (452–506)	466.0 (433–554)	0.172
Rrs	18.0 (11.5–31.5)	22.0 (14–28.2)	23.0 (14–24)	19.0 (13.5–22.3)	22.0 (16–24.8)	21.0 (16.5–26)	0.356

PIP: peak inspiratory pressure; Pplat: plateau pressure; Pmean: mean airway pressure; PEEP: positive end-expiratory pressure; RR: Respiratory rate; MVi: minute ventilation in inspiration; MVe: minute ventilation in expiration; VTi: Tidal volume in inspiration; VTe: Tidal volume in expiration; Rrs: maximum respiratory resistance.

**Table 3 pharmaceutics-13-00051-t003:** Blood gas parameters and PaO_2_/FiO_2_ at different time points.

Parameters	Zero Time	3rd h	24th h	*p* Value	*p* Value		
	Median (IQR)	Median (IQR)	Median (IQR)		Zero vs. 3rd h	Zero vs. 24th h	3rd h vs. 24th h
Physiology parameters							
HR	74.0 (57–103)	82.0 (64–97)	75.0 (61–92)	0.850			
MAP	76.0 (61–84)	72.0 (62–88)	66.0 (61–74)	0.643			
Blood gas parameters							
PH	7.45 (70–82.6)	7.42 (7.39–7.47)	7.44 (7.4–7.47)	0.746			
PaO_2_	74.4 (31.8–52.3)	80.8 (71.1–99.9)	84.8 (81–107.2)	0.078			
PaCO_2_	37.9 (22.8–30.6)	40.0 (35–54.4)	40.8 (32.7–44.5)	0.850			
HCO_3_^-^	27.4 (22.8–30.6)	26.6 (23.4–30.6)	26.7 (21.1–34.9)	0.423			
BEB	2.1 (−2.15–6.93)	1.6 (−1.88–6.73)	3.7 (0.8–7.95)	0.301			
FiO_2_	0.4 (0.4–0.4)	0.4 (0.3–0.4)	0.4 (0.3–0.4)	0.050			
PaO_2_/FiO_2_	191.0 (175–221)	221.75 (202–292)	246.7 (211.25–324)	0.012 *	0.407	0.009 **	0.407

HR: heart rate; MAP: mean arterial pressure; PaO_2_: Partial Pressure of Oxygen; PaCO_2_: partial pressure of carbon dioxide; HCO_3_−: bicarbonate; BEB: base excess; FiO2: The fraction of inspired oxygen; PaO_2_/FiO_2_: partial pressure of oxygen to fractional inspired oxygen. * represented *p* < 0.05; ** represented *p* < 0.01.

## Data Availability

The data presented in this study are available on request from the corresponding author. The data are not publicly available due to not mention about this issue in the informed consent.

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
