# Peer review of "Effect of Tiotropium Soft Mist Inhalers on Dynamic Changes in Lung Mechanics of Patients with Chronic Obstructive Pulmonary Disease Receiving Mechanical Ventilation: A Prospective Pilot Study"

_pharmaceutics, 2020, doi:10.3390/pharmaceutics13010051_

Round 1

Reviewer 1 Report

Congratulations for a well written interesting manuscript

Author Response

Thank you for your kindly review and suggestion.

Reviewer 2 Report

This study presents the effects of tiotropium bromide soft mist inhalers (SMIs) in patients with COPD receiving mechanical ventilation which is not explored sufficiently in clinic.

Although the parameters of lung mechanics did not reach statistical significance, trends of parameters were well discussed.

Following suggestions are recommended to improve the manuscript;

  • Font and symbol size in Figure, Table, and Foot note
  • Although the manuscript has a contribution on the pilot result to tiotropium bromide soft mist inhalers with mechanical ventilator, drug delivery efficiency is most related to aerosol particle size, velocity, output rate even the aerosol sprayed in the ventilation circuit. Therefore, SMI and its characteristics should be described in the method and discussion. It is also recommended to clarify where SMI was applied in the ventilation circuit.

Reviewer 3 Report

two minor observation

in the methods section please mention if you used tiotropium only one day or if for several days

please try to briefly describe which of the lung ventilation parameters are able to describe the rest hyperinflation (air  trapping) and so that their improvement also demonstrate a reduction of this hyperinflation as a result of tiotropium dosing.
